# Are better existing WASH practices in urban slums associated with a lower long-term risk of severe cholera? A prospective cohort study with 4 years of follow-up in Mirpur, Bangladesh

Sophie Kang ![ORCID],[1] Fahima Chowdhury,[2,3] Juyeon Park,[1] Tasnuva Ahmed,[2] Birkneh Tilahun Tadesse,[1] Md. Taufiqul Islam,[2] Deok Ryun Kim,[1] Justin Im,[1] Asma Binte Aziz,[1] Masuma Hoque,[2] Gideok Pak,[1] Farhana Khanam,[2] Faisal Ahmmed,[2] Xinxue Liu ![ORCID],[4] K Zaman ![ORCID],[2] Ashraful Islam Khan,[2] Jerome H Kim,[5] Florian Marks ![ORCID],[1,6] Firdausi Qadri,[2] John D Clemens[2,5,7]

SK and FC contributed equally.

SK and FC are joint first authors.

For numbered affiliations see end of article.

**Correspondence to**
Dr Sophie Kang;
sophie.kang@ivi.int

## ABSTRACT

**Objective** To investigate the association between existing household water quality, sanitation and hygiene (WASH) practices and severe cholera risk in a dense urban slum where cholera is highly endemic.

**Design, setting and participants** We assembled a large prospective cohort within a cluster randomised trial evaluating the effectiveness of oral cholera vaccine. Our dynamic cohort population (n=193 576) comprised individuals living in the 'non-intervention' clusters of the trial, and were followed over 4 years. This study was conducted in a dense urban slum community of Dhaka, Bangladesh and cholera surveillance was undertaken in 12 hospitals serving the study area.

**Primary outcome measure** First severe cholera episode detected during follow-up period.

**Methods** We applied a machine learning algorithm on a training subpopulation (n=96 943) to develop a binary ('better', 'not better') composite WASH variable predictive of severe cholera. The WASH rule was evaluated for performance in a separate validation subpopulation (n=96 633). Afterwards, we used Cox regression models to evaluate the association between 'better' WASH households and severe cholera risk over 4 years in the entire study population.

**Results** The 'better' WASH rule found that water quality and access were the most significant factors associated with severe cholera risk. Members of 'better' WASH households, constituting one-third of the population, had a 47% reduced risk of severe cholera (95% CI: 29 to 69; p<0.001), after adjusting for covariates. The protective association between living in a 'better' WASH household and severe cholera persisted in all age groups.

**Conclusions** Salutary existing household WASH practices were associated with a significantly reduced long-term risk of severe cholera in an urban slum of Dhaka. These findings suggest that WASH adaptations already practised in the community may be important for developing and implementing effective and sustainable cholera control programmes in similar settings.

## STRENGTHS AND LIMITATIONS OF THIS STUDY

⇒ We studied a cohort prospectively followed for 4 years, making this analysis one of the longest uninterrupted evaluations of the relationship between household water quality, sanitation and hygiene (WASH) and endemic cholera.

⇒ The multivariable WASH prediction rule was rigorously validated using a separate subpopulation and thus avoided overfitting to the training set.

⇒ We focused on the existing variability of household WASH within a dense slum community to demonstrate that there are salutary practices that may reduce the long-term risk of severe cholera.

⇒ Household WASH factors were only evaluated once, either at baseline or when participants entered the study area, and this status applied to the entire follow-up period.

⇒ The household WASH variables included in this study were collected in the context of an oral cholera vaccine trial and not optimised for describing WASH factors independently.

**Trial registration number** This article is a re-analysis of data from a cluster randomized trial; can be found on ClinicalTrials.gov NCT01339845

## INTRODUCTION

The low-income and middle-income world has seen a rapid expansion of urban areas due, in part, to influxes into squalid urban slums. With an estimated 55% of the world's population living in urban areas, one in three of those urban dwellers live in slum households.[1] Slum households are defined as those which lack one or more of the following conditions: access to improved water, access

to improved sanitation, sufficient living area and durability of housing.

In Bangladesh, urban dwellers currently account for 38% of the population and are expected to exceed 50% by 2030; furthermore, 47% of the urban population lives in slums, where residents are at increased risk for waterborne diseases, including cholera.[2 3] Cholera is a major cause of morbidity and mortality in Bangladesh.[4] Previously thought to be a rural disease in Bangladesh, cholera is now becoming a disease of cities and slums where living conditions create different challenges for cholera control from those encountered in rural settings.[5] Lessons learnt from rural areas, and particularly in epidemic situations, may not be applicable to the changing pattern of cholera endemicity in urban areas. Specific studies on how endemic cholera can be controlled in these urban slums are needed.

Exacerbating the urban cholera situation further is the role of climate change. Rising temperatures and increased precipitation associated with climate change are significant predictors of cholera incidence, with strong evidence in studies of Bangladesh where warmer, wetter conditions are associated with major cholera outbreaks.[6] Climate change effects in urban areas have added negative implications for water quality, and studies have found that increased rainfall intensity combined with impervious urban surfaces are significant predictors of sewer overflows that greatly impact water quality.[7 8]

Improvements of water quality, sanitation and hygiene (WASH) and oral cholera vaccine (OCV) are the major tools for the prevention of endemic cholera, including in urban slums. However, while WASH interventions are frequently employed to control cholera, evidence regarding their effectiveness is inconsistent and successful implementation may be stymied by limited cultural acceptability, low uptake and poor community acceptance.[9–12] Our cluster randomised trial (CRT) of OCV and WASH in an urban Dhaka slum, which failed to demonstrate that WASH added to protection against severe cholera by OCV, is illustrative.[13]

In this re-analysis of the CRT, we followed households in the 'non-intervention' arm to investigate how existing WASH practices and adaptations in the slums may be associated with lower severe cholera risk. We assessed long-term severe cholera risk over 4 years and sought to identify the salutary practices that might inform the development of future effective, acceptable and sustainable WASH interventions in cholera-endemic urban populations.

## METHODS

### Trial and population

In 2011, the International Centre for Diarrhoeal Disease Research, Bangladesh (icddr,b) conducted a CRT entitled 'Introduction of Cholera Vaccine in Bangladesh (ICVB)'[13] in six selected wards of Mirpur, Dhaka to evaluate the feasibility and effectiveness of OCV, both deployed alone and in conjunction with WASH interventions. Vaccination with Shanchol, a two-dose OCV, was carried out between 17 February and 6 April 2011.

Households were grouped into 90 geographic clusters with an average population of 2988 households per cluster (ranging from 2288 to 4299 households per cluster). Clusters were randomly assigned (1:1:1) to one of three arms: a two-dose regimen of OCV alone, OCV with a WASH intervention or no intervention (control).[13] The two doses of OCV were administered at a 14-day interval. The WASH intervention consisted of a behavioural change intervention, which focused on use of a household handwashing station and a chlorine dispenser for the treatment of household drinking water.[13] Healthy, non-pregnant individuals aged 1 year or older were eligible for vaccination in this CRT, and each cluster was separated from the adjoining cluster by at least a 30 m buffer area to minimise diffusion of the WASH messages between clusters.

### Demographic surveillance

The study population was characterised by a baseline demographic census and recurring census updates to surveille births, deaths and migrations in the community. The baseline demographic census was conducted before the start of the ICVB vaccination campaign and updated biannually. Verbal consent for participation in the surveillance was obtained and documented in a questionnaire at the time of the baseline census and at each biannual update. A 'household' was defined as persons sharing the same cooking pot.

In addition to the basic demographic information, household-level socioeconomic status, WASH data and geographical locations of each household were collected during the baseline census. For households that were not present at the baseline census, WASH characteristics were assessed at the first biannual census update they were captured in. Household WASH characteristics were not re-assessed for new births in a household already characterised during the study period. All individuals living in the study area were provided with a study identification card containing the unique participant identification number that was recorded in computerised study databases.

### Disease surveillance

Surveillance for cholera was conducted between April 2011 and November 2015 at 2 icddr,b hospitals and 10 hospitals serving the study area shown in figure 1. Study surveillance staff were present at each health facility throughout the day to facilitate reporting of diarrhoeal cases from the study area. Patients from the study area were identified in the treatment centres with their study identification card or by searching their identities in on-site computerised census database. Clinical examination was carried out by physicians, and designated study staff completed data forms and obtained faecal specimens after obtaining written informed consent.

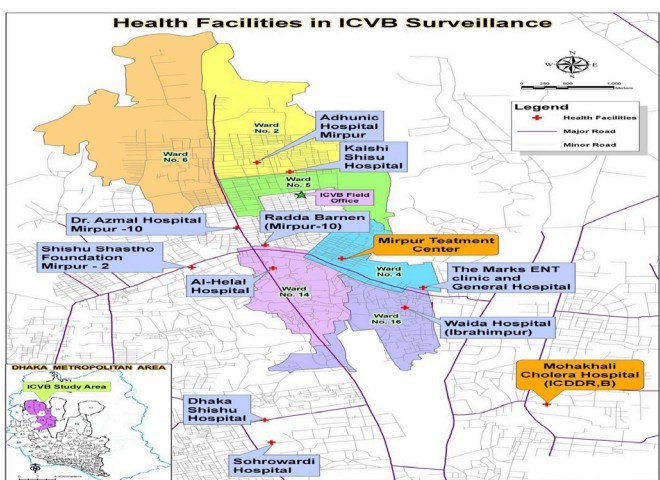

**Figure 1** Map of Introduction of Cholera Vaccine in Bangladesh (ICVB) study area and cholera surveillance treatment centres. icddr,b, International Centre for Diarrhoeal Disease Research, Bangladesh.

A diarrhoeal visit was defined as having three or more loose stools or, one to two or an indeterminate number of loose stools with evidence of dehydration, in the 24 hours before presentation.[13] If the date of discharge from an earlier diarrhoeal visit and the date of symptom onset for the subsequent diarrhoeal visit were within 7 days of one another, then both visits were considered part of the same diarrhoeal episode. The onset of a diarrhoeal episode was defined as starting on the day the patient first reported loose or liquid stools.

Faecal samples were examined for *Vibrio cholerae* O1 or O139 serogroups, biotype and Inaba and Ogawa serotypes using conventional methods.[13] A cholera episode was defined as a diarrhoeal episode in which a faecal specimen yielded *V. cholerae* O1 or O139, with no passage of bloody stools during the episode. Severely dehydrating cholera was defined by the presence of at least two of the following symptoms or signs of severe dehydration: sunken eyes, dry tongue, thirst, irritable condition, less active than usual along with inability to drink, skin pinch goes back slowly and low volume of radial pulse.[13] A severe cholera episode was one in which the patient exhibited severe dehydration during any visit of the episode. The primary outcome in this analysis was the first severe cholera episode detected during follow-up.

## Patient and public involvement

This analysis uses data that originate from the ICVB CRT conducted by the icddr,b in 2011. Given that 10 years have passed since the original study, the participants were not directly involved in developing or informing the design of the analysis described in this paper. That said, the original ICVB trial involved strong social mobilisation and community engagement to improve the conduct of the study.

The research questions addressed in the ICVB CRT were developed due to the pressing need to understand the impact of OCVs in urban Bangladesh. Advocacy

meetings with local government representatives, paediatric associations and non-governmental organisations were held in order to inform the design and conduct of the ICVB trial.

## ANALYTICAL APPROACH

### Population under follow-up

We considered a dynamic population for this analysis, which included the population present in the non-intervention arm at baseline and new entrants into the non-intervention study area during the study period. For those present at baseline, the start of follow-up was defined as the median date of first Shanchol dose in the nearest intervention cluster. For new residents, the start of follow-up was defined as either the date of birth or the date of migration into the study area. The end of follow-up was defined as either the end of surveillance, 4 years after baseline; date of death; date of migration out of the cluster or onset date of first severe cholera episode, whichever came first. Person-years of observation (PYO) were calculated from the sum of follow-up periods for individuals under analysis.

### Selection of WASH variables for analyses

We first examined the 10 household WASH variables ascertained in the demographic censuses and categorised each variable (shared toilet, drinking water source, distance to source of drinking water, drinking water treatment, toilet type, water availability, waste disposal location, hand washing water available, hand washing soap available and shared kitchen) into two categories: 'better' versus 'not better'. The categorisation of WASH variables was based on local context-informed judgement and the distribution of the study population into categories of the WASH variables, but without prior information on cholera incidence rates associated with each variable category.

We randomly divided the population of clusters of the 'non-intervention arm' of the trial into two subpopulations—50% of the households into a 'training' population and the other 50% into a 'validation' population. The training population was used to develop a recursive portioning tree to define the composite WASH variable, and the validation population was subsequently used to cross-validate the decision tree rule. We considered WASH variables associated with risk of severe cholera using a Cox proportional hazard regression model at p<0.2 using the training population. We verified that each selected independent variable fulfilled proportionality assumptions before inclusion into the model.

### Construction of decision tree to develop composite WASH variable

We developed a composite rule for existing 'better' versus 'not better' household WASH through a machine learning approach, recursive partitioning, to measure the association between WASH status and the incidence

of severe cholera up to 4 years of follow-up. To create a single, binary composite WASH variable predicting the occurrence of severe cholera, we considered variables that were associated with severe cholera in bivariate analyses.[14]

Given relatively few end points compared with the total number of individuals followed, we accounted for the imbalanced case distribution during construction of the decision tree. The decision tree was designed by assuming a 1:670 loss function for the cost of false positive and false negative classification, and by defining 300 as the minimum number of observations required in each terminal node. The number of cross-validations was 10. We ran the algorithm with the training subpopulation and subsequently pruned the tree by the minimal complexity parameter, corresponding to a minimum error with at least two terminal nodes to determine the optimal decision rule for predicting severe cholera.

With the selected tree, a receiver operating characteristic (ROC) curve was constructed in the training population to find the optimal cut-off probability of the composite WASH variable for predicting severe cholera.[15 16] This threshold was used to create a composite WASH variable with two categories, one for 'better' household WASH associated with a lower probability of developing severe cholera in household members, and the other for 'not better' household WASH, associated with a higher probability of developing severe cholera. We then evaluated the dichotomous WASH variable in an independent validation population to confirm that the prediction rule exhibited similar sensitivity and specificity for severe cholera in both populations.

### Protective association between WASH and severe cholera
Next, we measured the association between 'better' WASH household status and severe cholera in the entire population residing in non-intervention clusters. To evaluate this association, we analysed the time from start of follow-up to the first severe cholera case using the Cox proportional hazard regression model.

The model was adjusted for potential confounding covariates, including age in years at start of follow-up, sex and variables reflecting household socioeconomic status: monthly expenditure, house ownership, house having one room and house wall constructed by brick/cement. We introduced variables into the model by mixed stepwise selection, using a combination of forward and backward selection with the cut-off p value of 0.1 for both elimination and retention. HRs for severe cholera were estimated by exponentiating the coefficient for the composite WASH variable in models and protection was estimated as [(1−HR)×100%] with 95% CIs. Estimates were also adjusted for design effect of cluster randomisation of the study clusters.

To determine cluster-level 'better' WASH coverage, the person-years of observation of household members living in 'better' WASH households in the cluster were divided by the person-years of the entire population in the same cluster. The association between cluster-level WASH

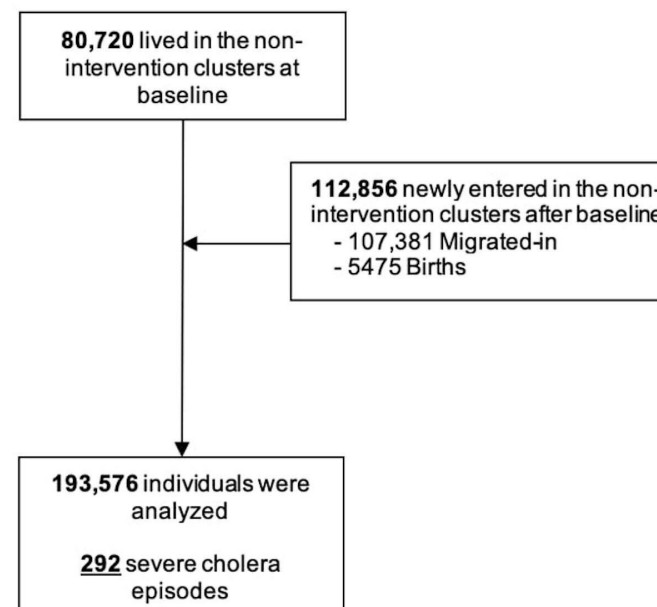

**Figure 2** Consolidated Standards of Reporting Trials—dynamic population during 4-year follow-up period.

coverage and incidence of severe cholera was assessed after adjustment for the same potential confounding variables in proportional hazard models using the same approach.

Statistical analyses were performed using R Studio analytical software for decision tree modelling (rpart package), tree plotting (rpart.plot package) and ROC curve illustration (pROC package). Other statistical analyses were performed using SAS V.9.4. All p values were two-sided.

## RESULTS
### Training and validation subpopulations
A total of 193 576 individuals in the non-intervention ICVB arm were included in the analysis, as shown in the Consolidated Standards of Reporting Trials diagram in figure 2. Of those, 80 720 individuals were present at baseline and 112 856 individuals were new entrants (107 381 in-migration individuals, 5475 births). During the 4 years of follow-up, a total of 292 severe cholera episodes were observed. The training set was composed of 96 943 individuals, 144 of whom developed severe cholera. The validation set was composed of 96 633 individuals, 148 of whom developed severe cholera. As shown in table 1, baseline characteristics of the training and validation subpopulations were broadly comparable in terms of mean age, sex ratio, average monthly household expenditure and other household characteristics.

### Rule development for composite WASH variable predicting severe cholera
A bivariate analysis for each variable in the training population was performed to measure associations of individual WASH-related variables with the risk of severe cholera

**Table 1** Baseline characteristics of the total, training and validation subpopulations

| | Total population | Training subpopulation | Validation subpopulation |
| --- | --- | --- | --- |
| | N=193 576 | N=96 943 | N=96 633 |
| Age in years—mean (SD) | 22.9 (15.4) | 22.9 (15.3) | 22.9 (15.4) |
| Gender: male—n (%) | 94 008 (48.6) | 47 365 (48.9) | 46 643 (48.3) |
| Monthly expenditure*—mean (SD) | 10 288.6 (5374.1) | 10 293.1 (4891.7) | 10 284.1 (5817.7) |
| Ownership: own house—n (%) | 28 677 (14.8) | 14 549 (15.0) | 14 128 (14.6) |
| House: having one room—n (%) | 165 215 (85.3) | 82 427 (85.0) | 82 788 (85.7) |
| Wall: brick/cement—n (%) | 139 860 (72.3) | 69 880 (72.1) | 69 980 (72.4) |

*Expenditure in Bangladeshi takas.
n, number of individuals.

(table 2). Drinking water source (51% reduction of risk; 95% CI: −20 to 80), distance to source of drinking water (39% reduction of risk ; 95% CI: 14 to 57) and drinking water treatment (36% reduction of risk; 95% CI: 11 to 54) were selected to determine the composite binary WASH variable. The resulting tree, shown in figure 3, found that treatment of drinking water was the dominant bifurcation; regardless of other WASH variables in the tree, not treating drinking water categorised the household as having 'not better' WASH.

### Performance of the composite WASH variable predicting severe cholera

In the training set, we found that an optimal cut-off value of 0.0012 for the composite WASH variable maximised the Youden index using the ROC curve, with an area under the curve (AUC) of 59% (95% CI: 55 to 63) (figure 4). Under this threshold, the rule predicted 123 true positives of 144 severe cholera episodes, for a sensitivity of 85% (95% CI: 85 to 86), and 28 709 true negatives among 96 799 persons without severe cholera, yielding a specificity of 30% (95% CI: 22 to 37). The composite WASH variable performed similarly when applied to the validation population, with a sensitivity of 82% (95% CI: 82 to 82) and specificity of 30% (95% CI: 22 to 37).

### Prediction of severe cholera incidence by household WASH status in the total population

We applied the WASH prediction rule to the total population residing in the non-intervention clusters to predict severe cholera episode risk. In the non-intervention arm, 29.7% of households were classified as having 'better' WASH and the remaining 70.3% were classified as having 'not better' WASH.

The incidence of severe cholera in all age groups living in the 'better' WASH households was 57 per 100 000 PYO, compared with 120 per 100 000 PYO in 'not better' WASH households, a 47% (95% CI: 29 to 61, p<0.001) reduced risk of severe cholera, after adjusting for covariates. A protective association between living in a 'better' WASH households and severe cholera risk was seen in all age groups (table 3). For individuals under the age of 15 years, living in a 'better' WASH households was associated

with having 64% (95% CI: 36 to 80; p<0.001) reduced risk of severe cholera. The protective association of 'better' WASH household status was somewhat lower in the 15+ year age group, where 'better' WASH was associated with a 43% (95% CI: 16 to 61; p=0.004) reduction in severe cholera risk.

There was a slight, non-significant negative association between rising proportion of 'better' WASH households in a cluster and incidence rate of severe cholera. In our model, for every 10% increase in proportion of 'better' WASH households in a cluster, individuals living in that cluster experienced 4% lower risk of severe cholera, although this relationship was not significant (HR=0.996 (95% CI: 0.988 to 1.005; p=0.40).

### DISCUSSION

Using a machine learning approach, we developed a composite WASH variable characterising households at baseline that predicted the risk of severe cholera risk for 4 years of follow-up of a cohort in a Dhaka slum. Water quality and access were the most significant factors associated with severe cholera risk.

This finding is consistent with previous conclusions in a systematic review of case-control studies by Wolfe *et al* that found strong associations between household water quality and lowered incidence of cholera.[17] In this review of mostly epidemic cholera contexts, the authors found that eight WASH-related risk factors, including unimproved water source, untreated water, unsafe water storage and transport, were consistently associated with higher odds of cholera. As such, the importance of water practices in our determination of 'better' WASH is in line with the prevailing understanding of cholera risk. What is new about our findings is that the protective relationships between household WASH and cholera also pertain to cholera of life-threatening severity, are sustained for at least 4 years after initial characterisation of household WASH at baseline and pertain to a densely populated, poor slum in which cholera is highly endemic.

In our analysis, we found that individuals living in 'better' WASH households had 47% (95% CI: 29 to 61)

**Table 2** Bivariate relationship of WASH variables with severe cholera risk in the training subpopulation

| WASH variable / Criteria for better WASH categorisation | Better WASH | | | | Not better WASH | | | | Protective association between variable and severe cholera | |
|---|---|---|---|---|---|---|---|---|---|---|
| | N | Cases | Person-years | IR (95% CI) Per 100 000/PYO | N | Cases | Person-years | IR (95% CI) Per 100 000/PYO | Crude (95% CI) | P value |
| Shared toilet *No* | 4620 | 9 | 10253 | 88 (46 to 169) | 92323 | 135 | 134076 | 101 (85 to 119) | 21 (−55 to 60) | 0.49 |
| Toilet type *Sanitary latrine with or without flush* | 80974 | 122 | 118747 | 103 (86 to 123) | 15969 | 22 | 25582 | 86 (57 to 131) | −24 (−95 to 21) | 0.36 |
| Drinking water source *Own tap/Well/Pump/Bottled or vendor water* | 4187 | 5 | 8993 | 56 (23 to 134) | 92756 | 139 | 135337 | 103 (87 to 121) | 51 (−20 to 80) | 0.12 |
| Water availability *Tap/Tube well/Well water is available all the time* | 67831 | 95 | 93293 | 102 (83 to 125) | 29112 | 49 | 51036 | 96 (73 to 127) | −15 (−64 to 19) | 0.43 |
| Distance to source of drinking water *Using median as cut-off* | 43806 | 47 | 64461 | 73 (55 to 97) | 53137 | 97 | 79869 | 121 (100 to 148) | 39 (14 to 57) | <0.01 |
| Drinking water treatment *Filtered/Boiled/Chemical treated* | 54275 | 60 | 77380 | 78 (60 to 100) | 42668 | 84 | 66950 | 125 (101 to 155) | 36 (11 to 54) | <0.01 |
| Waste disposal location *Fixed* | 80198 | 109 | 116056 | 94 (78 to 113) | 16745 | 35 | 28273 | 124 (89 to 172) | 20 (−17 to 46) | 0.25 |
| Hand washing water available* *Yes* | 93758 | 138 | 138159 | 100 (85 to 118) | 3185 | 6 | 6170 | 97 (44 to 216) | −11 (−152 to 51) | 0.8 |
| Hand washing soap available* *Yes* | 91514 | 133 | 135316 | 98 (83 to 117) | 5429 | 11 | 9014 | 122 (68 to 220) | 17 (−54 to 55) | 0.56 |
| Shared kitchen *No* | 86713 | 118 | 120182 | 98 (82 to 118) | 10230 | 26 | 24147 | 108 (73 to 158) | −5 (−62 to 32) | 0.83 |

*Indicates WASH condition observed directly by study team.

IR, incidence rate; n, number of individuals; PYO, person-years observation; WASH, water quality, sanitation and hygiene.

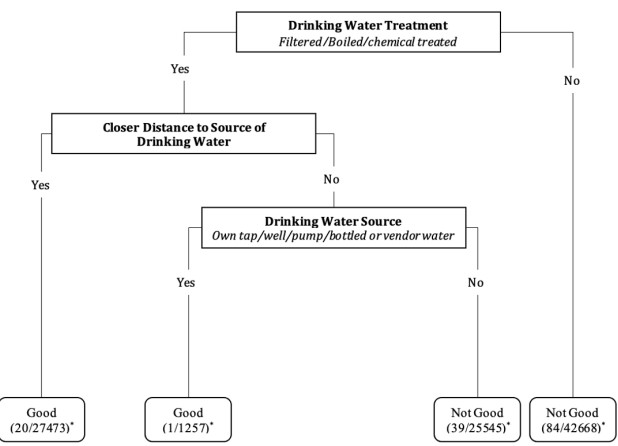

*Number of severe cholera cases / household population in the training subpopulation*

**Figure 3** Decision rule predicting severe cholera episode risk in the training subpopulation.

reduced risk of severe cholera compared with individuals living in 'not better' WASH households, after adjusting for age, gender and socioeconomic factors. This statistically significant protective association was demonstrated in all age groups examined, although those under the age of 15 years exhibited a greater degree of protection compared with those over the age of 15 years. This difference by age group may stem from the differences in WASH behaviours and exposure to cholera—where individuals aged 15 years and older are more likely to have many exposures outside of the household at school, work or play that could reduce the protective effects of living in a 'better' WASH home.

There are several strengths to our analysis that lend credibility to the findings. The analysis was evaluated in the context of a prospective CRT in a well-defined population with comprehensive follow-up for cholera detection. We followed this population for 4 years, making it one of the longest evaluations of the relationship between household WASH and severe cholera incidence, and thereby shedding light on the long-term durability of WASH

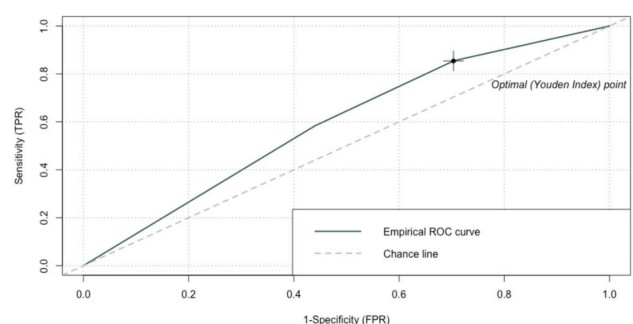

AUC = 59% (95% CI, 55-63); optimal cut-off probability using the Youden index = 0.0012

**Figure 4** Receiver operating characteristic (ROC) curve for the performance of the decision rule in the training subpopulation. AUC, area under the curve; FPR, false positive rate; TPR, true positive rate.

**Table 3** Protective association between 'better' WASH and severe cholera risk in the total study population

| Age at start of follow-up | 'Better' WASH | | | | 'Not better' WASH | | | | Protective association between 'better' WASH and severe cholera | | | |
|---|---|---|---|---|---|---|---|---|---|---|---|---|
| | N | Cases | Person-years | IR per 100 000/PYO (95% CI) | N | Cases | Person-years | IR per 100 000/ PYO (95% CI) | Crude (95% CI) | P value | Adjusted* (95% CI) | P value |
| <5 years | 7552 | 5 | 10518 | 48 (20 to 114) | 17149 | 42 | 24366 | 172 (127 to 233) | 72 (25 to 89) | 0.012 | 69 (15 to 88) | 0.022 |
| 5–14 years | 8861 | 4 | 14618 | 27 (10 to 73) | 25318 | 32 | 42389 | 75 (53 to 107) | 63 (6 to 85) | 0.037 | 63 (6 to 85) | 0.037 |
| 15+ years | 40983 | 39 | 59680 | 65 (48 to 89) | 93713 | 170 | 136945 | 124 (107 to 144) | 46 (20 to 64) | 0.002 | 43 (16 to 61) | 0.004 |
| All | 57396 | 48 | 84817 | 57 (43 to 75) | 136180 | 244 | 203701 | 120 (106 to 136) | 52 (34 to 65) | <0.001 | 47 (29 to 61) | <0.001 |

*Adjusted for design effect and selected covariates by stepwise selection using cut-off 0.1 for both of elimination and addition. For entire population, selected covariates were age, gender, home ownership and house having one room. For <5 years, no covariate was selected. For 5–14 years, selected covariate was home ownership. For 15+ years, selected covariates were age, gender and house having one room.
IR, incidence rate; n, number of individuals; PYO, person-years observation.

adaptations. Furthermore, during the development of the composite WASH rule we reduced bias by categorising WASH variables without any prior knowledge of cholera incidence rate associated with each component category. We also rigorously validated the WASH prediction rule by using a validation subpopulation to ensure that the rule was not overfitted to the training subpopulation.

Our analysis also has limitations. First, household WASH variables were only evaluated once and applied to the whole study period. As household WASH variables were more likely to have improved rather than regressed over time due to the overall secular improvement in socioeconomic conditions in Bangladesh, this misclassification, which would have affected both households classified at baseline as having 'better' WASH and those classified as having 'not better' WASH, would be expected to have led to more conservative estimates of protective associations with baseline WASH. Second, it should be noted that the household WASH variables included in this study were collected in the context of an OCV trial through a brief questionnaire designed to assess confounding variables in the evaluation of vaccine effectiveness. This approach may have led to loss of information, making our estimates of the WASH-severe cholera relationship conservative, and may also have identified variables that are 'proxies' for actual water-related factors that directly mediate a reduced risk of severe cholera. Consequently, the WASH variables included in the composite rule may best be interpreted as *markers* of these direct mediators. Nonetheless, because these variables predicted severe cholera risk independently of non-WASH socioeconomic factors, the analysis underscores the importance of water-related WASH adaptations in determining the risk of severe cholera in this setting. Finally, it might be queried whether the relationship between 'better WASH' in the household and a lower risk of severe cholera might reflect the possibility that the 'better WASH' households instituted home-based and clinic-based care for cholera early, and thereby forestalled progression to severe cholera. We think that this was an unlikely explanation for our findings because the study population was highly sensitised to home-based and clinic-based early treatment of watery diarrhoea, the population had very good access to clinics and hospitals in the proximities of their homes and all care for diarrhoea at these facilities was low cost or free of charge. Moreover, in analyses to be published elsewhere, we found that the household WASH prediction rule that we developed for severe cholera also strongly predicted the risk of all episodes of cholera seen at the treatment centres, regardless of severity.

Our findings indicate that there are existing culturally acceptable WASH improvements that may be impactful in controlling severe cholera in Mirpur, a dense slum population considered to have hyperendemic cholera. Past experiences in WASH intervention programmes have shown that achieving sustainable and effective WASH interventions can be challenging. For example, an evaluation of a household water treatment and handwashing campaign in rural Guatemala found that 3 years after the intervention there were no differences in handwashing behaviour, WASH conditions or prevalence of childhood diarrhoea in the community.[18] Other examples of poor long-term uptake and acceptability of WASH interventions are found in programmes implemented in India, Zambia and Kenya—all highlighting the difficulty of sustained behaviour adoption.[19–21] In light of this, understanding *existing* 'better' WASH households and behaviours in endemic slum communities can provide valuable lessons on designing feasible and sustainable WASH interventions.

Recent studies have also found that Mirpur, Bangladesh, the site for this study, has very high rates of microbiological proliferation and contamination in the municipal water supply,[22] likely due to antiquated and now porous water and sewage pipes, making achievable WASH improvements paramount to improving community health. Our analysis found that there are existing household water adaptations in Mirpur associated with significantly lowered severe cholera risk, despite contaminated municipal water. The fact that the WASH adaptations practised by 30% of the population were significantly predictive of lower severe cholera risk even within slum conditions speaks to the potential for existing knowledge to inform cholera control strategies.

Cholera transmission in urban slums may only intensify with the pressures of rapid urbanisation combined with climate change effects. Endemic cholera in urban areas is a reality in Bangladesh that faces challenges that are different from previously studied epidemic and rural cholera transmission. Closely examining how some urban households have already made WASH adaptations to reduce cholera risk may help design effective cholera control programmes that are sustainable and achievable in similar settings.

**Author affiliations**
[1]Epidemiology, Public Health, and Impact Unit, International Vaccine Institute, Gwanak-gu, The Republic of Korea
[2]Infectious Diseases Division, International Centre for Diarrhoeal Disease Research, Dhaka, Bangladesh
[3]Griffith University Menzies Health Institute Queensland, Nathan, Queensland, Australia
[4]Oxford Vaccine Group, Department of Paediatrics, University of Oxford, Oxford, UK
[5]International Vaccine Institute, Gwanak-gu, The Republic of Korea
[6]Department of Medicine, University of Cambridge, Cambridge, UK
[7]Fielding School of Public Health, University of California, Los Angeles, California, USA

**Correction notice** This article has been corrected since it first published. A joint first author and equally contributed statement have been added in the gutter section on the first page.

**Acknowledgements** Icddr,b is grateful to the Governments of Bangladesh, Canada, Sweden and the UK for providing unrestricted support. The authors would like to thank Dr Nigel McMillan (Griffith University) for his scientific and technical support in the manuscript.

**Contributors** JDC conceptualised the study and is the guarantor. SK and FC prepared the original draft. Data analysis was carried out by JP, FA and DRK. JDC, FQ, FM, JHK and AIK supervised the study. TA, BTT, TI, JI, ABA, MH, GDP, FK, XL and

KZ participated in interpreting the data. All authors reviewed and approved the final manuscript.

**Funding** This work was supported, in whole or in part, by the Bill & Melinda Gates Foundation (INV 025386). Under the grant conditions of the Foundation, a Creative Commons Attribution 4.0 Generic License has already been assigned to the Author Accepted Manuscript version that might arise from this submission.

**Disclaimer** The funders had no role in study design, data collection and analysis, decision to publish or preparation of the manuscript.

**Map disclaimer** The inclusion of any map (including the depiction of any boundaries therein), or of any geographic or locational reference, does not imply the expression of any opinion whatsoever on the part of BMJ concerning the legal status of any country, territory, jurisdiction or area or of its authorities. Any such expression remains solely that of the relevant source and is not endorsed by BMJ. Maps are provided without any warranty of any kind, either express or implied.

**Competing interests** None declared.

**Patient and public involvement** Patients and/or the public were not involved in the design, or conduct, or reporting, or dissemination plans of this research.

**Patient consent for publication** Not applicable.

**Ethics approval** This study is a re-analysis of data from a cluster randomised trial entitled 'Introduction of Cholera Vaccine in Bangladesh (ICVB)' conducted by the icddr,b in 2011. The ICVB trial received Institutional Review Board approval (icddr,b protocol number PR#17124) and Bangladesh National Research Ethics Committee approval. Participants gave informed consent to participate in the study before taking part.

**Provenance and peer review** Not commissioned; externally peer reviewed.

**Data availability statement** Data are currently available on reasonable request and will soon be made available in a public, open access data repository.

**ORCID iDs**
Sophie Kang http://orcid.org/0000-0001-9585-3556
Xinxue Liu http://orcid.org/0000-0003-1107-0365
K Zaman http://orcid.org/0000-0002-1982-6879
Florian Marks http://orcid.org/0000-0002-6043-7170

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
