## [Reviewer comments · BMJ Open]

ARTICLE DETAILS

TITLE (PROVISIONAL)	Are better existing WASH practices in urban slums associated with a lower long-term risk of severe cholera? A prospective cohort study with four years of follow-up in Mirpur, Bangladesh
AUTHORS	Kang, Sophie; Chowdhury, Fahima; Park, Juyeon; Ahmed, Tasnuva; Tadesse, Birkneh Tilahun; Islam, Md. Taufiqul; Kim, Deok Ryun; Im, Justin; Aziz, Asma; Hoque, Masuma; Pak, GiDeok; Khanam, Farhana; Ahmmed, Faisal; Liu, Xinxue; Zaman, K; Khan, Ashraful Islam; Kim, Jerome; Marks, Florian; Qadri, Firdausi; Clemens, John

VERSION 1 – REVIEW

REVIEWER	Rahman, Aminur International Centre for Diarrhoeal Disease Research, Health Systems and Population Studies Division (HSPSD)
REVIEW RETURNED	08-Mar-2022

GENERAL COMMENTS	Title: Should include place, person and time in secular form. Need to revise Abstract Method: Line 30, Surveillance ward came abruptly, need a link sentence/s for that. Need to describe a bit about the treatment centers in the study area like; distribution, public or others type, staffs pattern etc. As this a four-years cohort, so the study participants might receive information on better WASH practices from other sources which might be a confounding factors, may distort the result and limitation of the study (information contamination)
--

REVIEWER	Bwire, Godfrey Republic of Uganda Ministry of Health, Intergrated Epidemiology, Surveillance and Public Health Emergencies
REVIEW RETURNED	21-Mar-2022

GENERAL COMMENTS	General comments The authors, Sophie Kang et al, studied the impact of WASH interventions on severe cholera in Mirpur, Dhaka and found WASH to significantly reduce the long term risk of severe cholera. This is an interesting study since it is on cholera, Cholera is an important public health issue that is a major cause of mortality and morbidity in developing world. The manuscript is well written and the authors used robust methodology to show how WASH could prevent severe cholera in endemic slum setting of Bangladesh. However, it is useful to note that prevention of diarrheal, cholera inclusive using WASH is a well-known fact (PMCID: PMC6073794, PMC2393264). Also, before a case of cholera progress to severe, it
---

	passes through mild form (cholera, no dehydration) and moderate form (cholera, moderate dehydration). Assuming that all other factors are constant, the progression from mild and moderate to severe form largely depend on the case management for cholera (https://apps.who.int/iris/bitstream/handle/10665/36837/924154449X.pdf ; PMID: 28957470, DOI: 10.1093/trstmh/trx041,) and where the case management is good (early diagnosis of cholera and initiation of appropriate rehydration therapy with or without antibiotic treatment) the severe form of cholera is prevented (PMID: 28957470, DOI: 10.1093/trstmh/trx041,). Specific comments In this study, the authors focus on severe cases and leave out the mild and moderate cases. They also, do not consider case management in prevention of severe diarrhea. This is major flaw that weaken this study. Therefore, the author will need to address the following; Major issues  1. The authors studied severe cholera, however in the title and in several section of the manuscript they keep on referring to cholera without differentiation which is prone to confuse the readers. Therefore, since they do not differentiate in most of the manuscript the authors need to show the impact on other cholera classification as well. Otherwise, they should revise the title and manuscript in line with the variable (Severe cholera) that was studied. 2. Progression of cholera from mild or moderate cholera to severe cholera depends largely on case management provided (time of diagnosis plus appropriate rehydration therapy and nutrition). In this study the authors are silent on the role of case management in prevention of severe cholera. Could be that the severe cases noted in this study were due to inadequate case management or lack of it? 3. The impact of WASH on diarrheal including cholera is not new (PMC5688426 [doi: 10.1186/s12889-017-4746-1], PMID: PMC2393264 [https://www.ncbi.nlm.nih.gov/pmc/articles/PMC2393264/], PMID: 7036486 [doi: 10.1016/S0140-6736(02)11403-6.], PMID: PMC2366240 [https://www.ncbi.nlm.nih.gov/pmc/articles/PMC2366240/]). These effect of WASH that is known is none selective and works for all classes of diarrhea (mild, moderate and severe). Therefore, It could be more exciting to readers of this paper if the authors could clearly show what is new in this manuscript that has not been covered by the earlier studies. 4. From the method section, page 8, lines 3-10 it very clear that the authors carried out the study in only one area (urban slum community of Dhaka, Bangladesh). However, they base on this single site to generalize their findings as shown in the title and in the conclusion Page 2, lines 45-52, “ These findings suggest that salutary WASH practices can significantly reduce long-term risk of severe cholera even in highly endemic areas, and future interventions should look to these culturally acceptable WASH practices when designing sustainable cholera programs” and in discussion Page 16 line 11-16, “Our findings indicate that there are existing culturally acceptable WASH improvements that can be impactful in controlling cholera in a dense slum population considered to have hyperendemic cholera”. The authors need to review this blanket generalization since other studies have shown contrasting epidemiology of cholera in Asia and Africa (PMID: PMC8687066, DOI: 10.1093/infdis/jiab440).
--	--

VERSION 1 – AUTHOR RESPONSE

Reviewer 1: Dr. Aminur Rahman, International Centre for Diarrhoeal Disease Research

Comments to the Author:

Title: Should include place, person, and time in secular form. Need to revise Abstract

Response: Thank you for the pertinent suggestion, we have revised the title to include this information and hope there is now improved clarity.

Revised title: “Are better existing WASH practices in urban slums associated with a lower long-term risk of severe cholera? A prospective cohort study with four years of follow-up in Mirpur, Bangladesh”

The abstract has also been amended to more clearly summarize the analysis methods and conclusion on page 2, lines 15-30. The changes have been highlighted in the revised manuscript.

Comment: Method: Line 30, Surveillance ward came abruptly, need a link sentence/s for that. Need to describe a bit about the treatment centers in the study area like; distribution, public or another type, staff's pattern, etc.

Response: The reviewer's recommendations are well-taken. The methods section has been revised to improve the transition to the demographic surveillance description with an introductory sentence on page 5, line 16: “The study population was characterized with a baseline demographic census and recurring census updates to surveille births, deaths, and migrations in the community.”

We have also included further details on cholera treatment centers on page 5 by adding a map of the locations of the surveillance centers as Figure 1.

Comment: As this is a four-year cohort, the study participants might receive information on better WASH practices from other sources which might be a confounding factor, and may distort the result and limitation of the study (information contamination)

Response: We appreciate this observation. The only WASH intervention taking place in the area was that provided to the OCV plus WASH clusters in the Introduction of Cholera Vaccine into Bangladesh (ICVB) trial, for which the control clusters were used in our present analysis. However, it is very unlikely that the WASH intervention for the trial had an impact on the control clusters, because the WASH intervention had no impact on the incidence of severe cholera in the OCV plus WASH clusters [Qadri et al, 2015; PMID: 26164097]. As well, the clusters for the trial were separated by 30 m buffer zones to minimize this kind of diffusion. Finally, in our study, we analyzed only baseline WASH practices. If later improved WASH behaviors and practices had arisen in the control cluster households during the four years of follow-up, they would have had to have occurred selectively in households characterized as having “Better WASH” at baseline to have explained our findings, which seems very unlikely.

In our discussion (Page 10 line 32) we note: 'As household WASH variables were more likely to have improved rather than regressed over time due to the overall secular improvement in socioeconomic conditions in Bangladesh, this misclassification, which would have affected both households classified at baseline as having "Better" WASH and those classified as having "Not Better" WASH, would be expected to have led to more conservative estimates of protective associations with baseline WASH.'

Reviewer 2: Dr. Godfrey Bwire, Republic of Uganda Ministry of Health, Makerere University College of Health Sciences

Comments to the Author:

General comments

The authors, Sophie Kang et al, studied the impact of WASH interventions on severe cholera in Mirpur, Dhaka, and found WASH to significantly reduce the long-term risk of severe cholera. This is an interesting study since it is on cholera, cholera is an important public health issue that is a major cause of mortality and morbidity in the developing world. The manuscript is well written and the authors used robust methodology to show how WASH could prevent severe cholera in the endemic slum setting of Bangladesh.

However, it is useful to note that the prevention of diarrheal, and cholera inclusive using WASH is a well-known fact (PMCID: PMC6073794, PMC2393264). Also, before a case of cholera progress to severe, it passes through the mild form (cholera, no dehydration) and moderate form (cholera, moderate dehydration). Assuming that all other factors are constant, the progression from mild and moderate to severe form largely depend on the case management for cholera (<https://apps.who.int/iris/bitstream/handle/10665/36837/924154449X.pdf> ; PMID: 28957470, DOI: 10.1093/trstmh/trx041,) and where the case management is good (early diagnosis of cholera and initiation of appropriate rehydration therapy with or without antibiotic treatment) the severe form of cholera is prevented (PMID: 28957470, DOI: 10.1093/trstmh/trx041,).

Specific comments

In this study, the authors focus on severe cases and leave out mild and moderate cases. They also, do not consider case management in the prevention of severe diarrhea. This is a major flaw that weakens this study. Therefore, the author will need to address the following;

Major issues

Comment: 1. The authors studied severe cholera, however, in the title and in several sections of the manuscript they keep on referring to cholera without differentiation which is prone to confuse the readers. Therefore, since they do not differentiate in most of the manuscripts the authors need to show the impact on other cholera classifications as well. Otherwise, they should revise the title and manuscript in line with the variable (Severe cholera) that was studied.

Response: Thank you for the astute observation. We have revised the title and manuscript body to reflect the focus on severe cholera rather than all cholera episodes.

Revised title: Are better existing WASH practices in urban slums associated with a lower long-term risk of severe cholera? A prospective cohort study with four years of follow-up in Mirpur, Bangladesh

Comment: Progression of cholera from mild or moderate cholera to severe cholera depends largely on case management provided (time of diagnosis plus appropriate rehydration therapy and nutrition). In this study, the authors are silent on the role of case management in the prevention of severe cholera. Could be that the severe cases noted in this study were due to inadequate case management or lack of it?

Response: The reviewer's query is well noted and the role of case management is certainly relevant to cholera severity. We now provide the following text in the Discussion (Page 11; lines 7-16) to address this concern:

"It might be queried whether the relationship between "Better WASH" in the household and a lower risk of severe cholera might reflect the possibility that the "Better WASH" households instituted home-based and clinic-based care for cholera early, and thereby forestalled progression to severe cholera. We think that this was unlikely because the study population was highly sensitized to home- and clinic-based early treatment of watery diarrhea, because the population had very good access to clinics and hospitals in the proximities of their homes, and because all care for diarrhea at these facilities was low cost or free of charge. Moreover, in analyses to be published elsewhere, we found that the WASH prediction rule for severe cholera also strongly predicted the risk of all episodes of cholera seen at the treatment centers."

Comment: The impact of WASH on diarrheal including cholera is not new (PMC5688426 [doi: 10.1186/s12889-017-4746-1], PMID: 7036486 [doi: 10.1016/S0140-6736(02)11403-6.], PMID: PMC2366240 [https://www.ncbi.nlm.nih.gov/pmc/articles/PMC2366240/]. These effect of WASH that is known is none selective and works for all classes of diarrhea (mild, moderate, and severe). Therefore, It could be more exciting to readers of this paper if the authors could clearly show what is new in this manuscript that has not been covered by the earlier studies.

Response: We now clearly state in the discussion on Page 10, lines 7-11: "What is new about our findings is that the protective relationships between household WASH and cholera also pertain to cholera of life-threatening severity, are sustained for at least four years after initial characterization of household WASH at baseline, and pertain to a densely populated, poor slum in which cholera is highly endemic."

Comment: From the method section, page 8, lines 3-10 it is very clear that the authors carried out the study in only one area (the urban slum community of Dhaka, Bangladesh). However, they base on this single site to generalize their findings as shown in the title and in the conclusion on Page 2, lines 45-52, " These findings suggest that salutary WASH practices can significantly reduce the long-term risk of severe cholera even in highly endemic areas, and future interventions should look to these culturally acceptable WASH practices when designing sustainable cholera programs" and in discussion Page 16 line 11-16, "Our findings indicate that there are existing culturally acceptable WASH improvements that can be impactful in controlling cholera in a dense slum population considered to have hyperendemic cholera". The authors need to review this blanket generalization since other studies have shown contrasting epidemiology of cholera in Asia and Africa (PMID:

Response: Thank you for the astute comment, we certainly agree that such blanket generalizations cannot be claimed from the results of this analysis. We have revised the language in the abstract conclusion and discussion to more accurately represent our findings and avoid oversimplification.

In the abstract conclusion (Page 2, lines 28-31), we revised the wording as: “Salutary existing household WASH practices were associated with a significantly reduced long-term risk of severe cholera in an urban slum of Dhaka. These findings suggest that WASH adaptations already practiced in the community **may** be important for developing and implementing effective and sustainable cholera control programs **in similar settings.**”

Also, in the discussion (Page 11, line 22), we revised the findings to be more specific to the study context: “Our findings indicate that there are existing culturally acceptable WASH improvements that **may** be impactful in controlling **severe** cholera in **Mirpur...**”

We understand that the results are specific to the study area in Dhaka, however, we do believe that the strategy of considering existing, culturally acceptable WASH practices may be relevant to many other cholera endemic contexts.

Reviewer: 1

Competing interests of Reviewer: I have no competing of interest

Reviewer: 2

Competing interests of Reviewer: None

VERSION 2 – REVIEW

REVIEWER	Bwire, Godfrey Republic of Uganda Ministry of Health, Intergrated Epidemiology, Surveillance and Public Health Emergencies
REVIEW RETURNED	02-Jun-2022

GENERAL COMMENTS	General overview I am happy with the revised manuscript and I commend the authors for the extra efforts. However, there is one essential issue that is now clearer and that is the definition of diarrhea, an important symptom for cholera disease. The definition in this paper is not according to that of the World Health Organization and in many other clinical books. Specific comment Page 10, Lines 13-15, the authors use a new definition of diarrhea, “A diarrheal visit was defined as having 3 or more loose stools or, 1-2 or an indeterminate number of loose stools with evidence of dehydration, in the 24 hours before presentation”. This “new definition of diarrhea” that ignore the frequency of loose stool represents new study findings that would attract the attention of many researchers and policy-makers who for a long time have
--

	defined diarrhea as per the World Health Organization definition (https://www.who.int/news-room/fact-sheets/detail/diarrhoeal-disease). In this definition, Diarrhea is defined as the passage of three or more loose or liquid stools per day (or more frequent passage than is normal for the individual). Frequent passing of formed stools is not diarrhea, nor is the passing of loose, "pasty" stools by breastfed babies. The WHO definition is widely accepted in many leading books such as John Malcolm Dowling CFY. Communicable Diseases Common in Developing Countries: Stopping the global epidemics of HIV/AIDS, Tuberculosis, Malaria and diarrhea. Encyclopedia of Human Services and Diversity. 2014. Available: https://www.pdfdrive.com/communicable-diseases-in-developing-countries-stopping-the-global-epidemics-of-hiv-aids-tuberculosis-malaria-and-diarrhea-d167120773.html I note that the same definition was used in the earlier study by Firdausi et al, DOI:https://doi.org/10.1016/S0140-6736(15)61140-0 on the same topic. However, I think that without clear justification, the authors of this manuscript should not use the same definition since it is not according to that by WHO and by other literature. Also, could it be that the observed association of severe symptoms with WASH was a result of this new diarrhea definition? Therefore, the authors will need to elaborate on the added value of using the new diarrhea definition as opposed to the existing one by the WHO and other authors.
--	---

VERSION 2 – AUTHOR RESPONSE

satisfactory. In response to the reviewer’s comment on diarrheal definition, we would like to provide further context.

The definition for ‘diarrheal visit’ in our analysis was used to help define “cholera” detected in treatment settings in the original, published OCV cluster randomized trial reanalyzed in our paper, and has been used in multiple trials of killed OCVs over the past several decades (Ali et al, 2021; doi: 10.1016/S1473-3099(20)30781-7). It was informed by the WHO criteria from 2005, “The treatment of diarrhoea : a manual for physicians and other senior health workers, 4th rev. World Health Organization.” (<https://apps.who.int/iris/handle/10665/43209?locale-attribute=en&>), but adapted for the detection of cholera among patients seeking care for diarrhea with the additional requirement that there be fecal culture confirmation of cholera. In contrast, the WHO criteria were primarily developed to assist in the determination of diarrhea at the household level without requiring fecal isolation of pathogens. We therefore believe that the definition of diarrhea used to help define cholera in our study was appropriate.